# Association between hematocrit and the 30-day mortality of patients with sepsis: A retrospective analysis based on the large-scale clinical database MIMIC-IV

**Mengdi Luo**[1], **Yang Chen**[1], **Yuan Cheng**[1], **Na Li**[2], **He Qing**[1] *

**1** ICU, Affiliated Hospital of Southwest Jiaotong University, The Third People's Hospital of Chengdu, Chengdu, Sichuan, China, **2** Southwest Jiaotong University, Chengdu, Sichuan, China

* kk555888@126.com

**Data Availability Statement:** All datasets supporting the conclusions of this study are

## Abstract

This research sought to ascertain the relationship between hematocrit (HCT) and mortality in patients with sepsis. Methods: A retrospective analysis was conducted on the clinical data of septic patients who were hospitalized between 2008 and 2019 in an advanced academic medical center in Boston, Massachusetts, registered in the Medical Information Mart for Intensive Care IV (MIMIC-IV) database, We analyzed basic information including gender, age, race, and types of the first admission, laboratory indicators including HCT, platelets, white blood cells, albumin, bilirubin, hemoglobin, and serum creatinine, and 30-day mortality. A Cox proportional hazards regression model was utilized to analyze the relationship between HCT and 30-day mortality in patients with sepsis. Results: This research recruited 2057 patients who met the research requirements from 2008 to 2019. According to the HCT level, it was classified into the low HCT level, the regular HCT level, and the high HCT level. The 30-day mortality rate was 62.6%, 27.5%, and 9.9% for patients with the low HCT level, the regular HCT level, and the high HCT level, respectively ($p < 0.05$). The multivariate Cox proportional hazard regression model analysis displayed that compared with patients with the regular HCT level, the 30-day mortality of patients with the low HCT level increased by 58.9% (hazard ratio = 1.589, 95% confidence interval (CI) = 1.009–2.979, $p < 0.05$). Conclusion: The low HCT level is an independent risk factor for the increase of the 30-day mortality in patients with sepsis and can be used as a significant predictor of the clinical outcome of sepsis.

## Introduction

Sepsis is a lethal syndrome of physiologic, pathologic, and biochemical abnormalities induced by infection, which is one of the major global public health concerns [1]. Although recently there has been extensive researches demonstrating the mechanism and treatment of sepsis, sepsis is still the principle cause of death in intensive care patients worldwide [2]. There are a

obtained from the MIMIC-III database (web site: https://mimic.physionet.org/).

**Funding:** This study was supported by (the Chengdu Science and Technology Bureau (No.2017FZ0058)) and (the Health Department of Sichuan Province (No.17PJ474)).The funders had no role in study design, data collection and analysis, decision to publish, or preparation of the manuscrip.

**Competing interests:** No

variety of scores for the diagnosis of sepsis that can be evaluated, but there is still a lack of valuable indicators for the study on prognostic factors [3]. Anemia is one of the risk factors for death resulting from sepsis and septic shock, and hematocrit (HCT) is the percentage of red blood cells in the volume of the whole blood, which is one of the critical biomarkers for the diagnosis of anemia [4].

HCT can be utilized as a critical prognostic biomarker in several cancers, including lung cancer, renal carcinoma, and epithelial ovarian cancer. However, little is known about the prognostic value of HCT for patients with sepsis in surgical intensive care units (ICUs) [5]. At present, clinical researches have mainly focused on the relationship of anemia indicators such as the red blood cell distribution width (RDW) and platelets with the prognosis of sepsis [6]. In spite of limited researches on the impacts of HCT on the prognosis of patients with sepsis, a few studies are limited to the evaluation of anemia before sepsis surgery [7]. Accordingly, the HCT level of first admission patients was employed in our research to dissect out whether it influenced the prognosis of sepsis, thus helping doctors assess the condition in a timely manner and provide a basis for subsequent prognostic measures.

## Method

### Data source

All data used in the study was extracted from the MIMIC-IV (v1.0) database, which is the hospitalization information of patients admitted to the Higher Medical Center in Boston, Massachusetts, the USA from 2008 to 2019 to conduct a retrospective study. This database is a relational database containing the actual hospitalizations of hospitalized patients in an advanced academic medical center in Boston, Massachusetts, USA. The MIMIC-IV database is based on the success of MIMIC-III [8]. It integrates improvements to the deficiencies of the MIMIC-III database, including laboratory measurements, medications, recorded vital signs, SOFA score, SAPS II score, etc.

### Study population and data extraction

PgAdmin4 has used run structure query language (SQL) and extracted data from the MIMIC IV database. Patients are identified by ICD-9 codes and extracted from the database [9]. The extracted variables included age, gender, admission type, comorbidities, laboratory parameters, severity score, time of entering and leaving the hospital, time of entering and leaving the ICU, and date of death. Laboratory parameters include hematocrit, albumin, lactate, bilirubin, hemoglobin, red blood cell distribution width, platelet, serum creatinine, potassium, etc. All laboratory data were extracted from the data generated within the first 24 hours after the patient entered the ICU (i.e., the baseline value). Data in this retrospective analysis is accurate medical data, accessed free of charge. The project has been approved by the Institutional Review Board of the Massachusetts Institute of Technology and BIDMC and has received an informed consent exemption. The author has the right to use and download the database through the Protecting Human Research Participants exam.

### Patient population

Adult patients (age ≥18 years) admitted to ICU (internal medicine ICU, surgical ICU) who had the exact hematocrit index and corresponding ID within 24 hours after admission to the ICU. Patients who lack corresponding data will be excluded. Sepsis criteria: This study uses patients diagnosed with sepsis in the database and meets sepsis-3 to define the sepsis cohort.

## Statistical analysis

Analyses were performed using Stata 16. Baseline characteristics of all patients were stratified according to the quartiles of hematocrit (HCT) and divided into three groups: low hct level (male≤42%, Female≤37%), regular hct level (male 42%~49%, female 37%~44%), and high hct level (male≥49%, female≥44%) [4]. Continuous variables were presented as median, and categorical variables were presented as a number. Nonparametric Wilcoxon tests or Kruskal-Wallis tests were used to compare continuous variables between different groups. Categorical variables between the groups were compared using the X2 test and Fisher exact test. A P value < 0.05 between the two groups was considered a significant difference. A Multivariate Cox proportional hazard model was constructed to determine the independent effects of three groups on 30-day mortality. Variables with P <0.05 in univariate analysis were further included in the multivariate Cox proportional hazard model. We compared the survival rates using log-rank tests and presented the results as Kaplan-Meier curves. A Kaplan-Meier survival curve was constructed to compare the 30-day mortality of the three groups of hct levels. P<0.05 indicates that the difference is statistically significant.

## Result

### Population and baseline characteristics

A total of 2057 patients with sepsis 3.0 in the MIMICIV database were included which have complete data of hematocrit level within 24 hours after ICU admission. All baseline characteristics are summarized in Table 1. Differences in age, laboratory parameters, comorbidities, and scores between the two groups were statistically significant. HCT level was divided into three groups. Variables with missing data are relatively common in the MIMIC IV database. The number of deaths within 30 days after admission to the ICU was 911. Compared with the 30-day survival group, the death group was older, and the levels of hematocrit, albumin, platelets, and hemoglobin were lower than those in the survival group (P<0.05).

### Association of Hematocrit level with 30-day outcomes

The 30-day all-cause mortality rate of patients was 44.2%, As shown in Table 2, compared with the survival group, the proportion of patients with low hct levels in the 30-day mortality group was significantly increased, and the proportion of the regular group and the high hct level group gradually decreased. The difference was statistically significant (P<0.05).

### Hematocrit is an independent prognostic predictor in sepsis patients

Survival analysis was conducted to explore the impact of hct level on 30d mortality. Notably, from the previous analysis, we know that patients in the lower hct level had worse survival rates. Basic demographics and laboratory parameters for the prediction of 30-day mortality were investigated using a univariate Cox analysis regression model. Variables include age, low hct levels, albumin, platelets, bilirubin, red blood cell distribution width, lactate, Scr, SBP, DBP MAP, SAPS II score, hypertension, heart failure, and mechanical ventilation are all statistically significant (p<0.05). Adjust the univariate analysis for the potential confounding factors associated with 30-day mortality in patients with sepsis, And then, the 30-day mortality was assessed with a multivariable Cox proportional regression model. According to the results, the low hct level remained an independent prognostic factor for sepsis (P<0,05). Compared with the regular hct level, the 30-day mortality risk of the low hct level is increased, and the difference is statistically significant [HR = 1.589 .95%CI1.099–2.297, P<0.05], although the risk of

**Table 1. Characteristics of the patients with sepsis.**

| Variables | 30-day mortality, n = 911 1 | 30-day survival, n = 1146 0 | P |
|---|---|---|---|
| Age, years | 70[59,81] | 66[54,79] | <0.001 |
| Female, n (%) | 418 (45.88) | 519 (45.29) | 0.788 |
| First care unit, n (%) | | | 0.035 |
| MICU | 294 (32.34) | 398(34.88) | |
| MICU/SICU | 254 (27.94) | 342 (29.97) | |
| SICU | 216 (23.76) | 263 (23.05) | |
| Others | 145 (15,95) | 138 (12.09) | |
| Mechanical Ventilation, n (%) | 247[27.11] | 232[20.24] | <0.001 |
| Comorbidities, n (%) | | | |
| Hypertension, n (%) | 106 (11.64) | 193 (16.94) | <0.001 |
| COPD, n (%) | 91 (9.99) | 93 (8.12) | 0.139 |
| Diabetes, n (%) | 46 (5.05) | 59 (5.15) | <0.001 |
| Respiratory failure, n (%) | 374 (41.05) | 309 (26.96) | 0.014 |
| CHF, n (%) | 42 (4.61) | 30 (2.62) | <0.001 |
| Vital signs | | | |
| SBP, mmHg | 113.41[105.0,119.2] | 116.3[110.3,119.5] | <0.001 |
| DBP, mmHg | 57.0[51.0,60.3] | 58.4[55.8,61.7] | <0.001 |
| Heart rate, bmp | 94.0[76.0,120.0] | 94.0[71.0,120.0] | 0.093 |
| MAP, mmHg | 72.0[63.0,79.5] | 76.8[70.0,80.0] | <0.001 |
| Laboratory parameters | | | |
| Albumin, g/dl | 3.0[2.5,3.6] | 3.2[2.7,3.7] | <0.001 |
| White blood cell,$10^9$/L | 14.2[10.8,17.9] | 14.2[11.3,18.0] | 0.507 |
| Hematocrit,% | 35.0[30.0.39.9] | 35,9[31.0,40.8] | <0.001 |
| Platelet,$10^9$/L | 153.0[91.0.230.0] | 181.0[125.0,255.0] | <0.001 |
| Bilirubin, mg/dL | 1.3[0.6,3.1] | 0.9[0.4,2.8] | <0.001 |
| Hemoglobin, g/dL | 9.6[9.4,9.9] | 9.6[9.5,9.8] | 0.037 |
| RDW,% | 14.6[13.6,16.2] | 13.9[13.0,15.1] | <0.001 |
| Scr, mg/dl | 1.6[1.0,2.6] | 1.3[0.6,2.1] | <0.001 |
| Lactate, mmol/L | 2.5[1.6,4.6] | 1.9[1.3,2.8] | <0.001 |
| Sodium, mEq/L | 138.0[134.0,142.0] | 139[141.0,135.0] | 0.186 |
| Potassium, mEq/L | 4.2[3.7,4.8] | 4.0[3.6,4.5] | <0.001 |
| Severity scores | | | |
| SOFA | 6.0[3.0,9.0] | 4.0[3.0,6.0] | <0.001 |
| SAPSII | 52.0[42.0,63.0] | 42.0[33.0,52.0] | <0.001 |

*Data are expressed as median (IQR), or n (%). Analysis of variance (or the Kruskal-Wallis test) and Chi-square (or Fisher's exact) tests were used for comparisons among groups. Statistical significance (P<0.05).

ICU, intensive care unit; SICU, Surgical Intensive Care Unit; MICU, Medicine Intensive Care Unit; COPD, Chronic obstructive pulmonary disease; CHF, Congestive heart failure; SBP, Systolic blood pressure; DBP, Diastolic blood pressure; MAP, mean arterial pressure; RDW, Red blood cell distribution width; Scr, Serum creatinine; SOFA, Sequential Organ Failure Assessment; SAPSII, Simplified Acute Physiology Scores II.

**Table 2. Outcomes of sepsis patients according to the hematocrit level.**

| | Low hematocrit level | Regular hematocrit level | High hematocrit level | P |
|---|---|---|---|---|
| 30-day mortality, n (%) | 569(62.63) | 251(27.55) | 91(9.99) | <0.05 |
| 30-day survival, n (%) | 653(56.98) | 358(31.24) | 135(11.78) | <0.05 |
| Total (n = 2057), n (%) | 1222(59.41) | 609(29.61) | 226(10.99 | <0.05 |

**Table 3. Univariate and multivariable analyses for the relationship between the candidate risk factors and 30- day mortality in the primary cohort.**

| Variables | Univariate model | | | Multivariable model | | |
|---|---|---|---|---|---|---|
| | HR | 95%CI | P | HR | 95%CI | P |
| Age | 1.018 | 1.014  1.023 | <0.001 | 1.016 | 1.011  1.022 | <0.001 |
| Female | 1.105 | 0.970  1.259 | 0.133 | | | |
| MAP | 0.996 | 0.994  0.998 | <0.001 | 0.998 | 0.995  0.999 | 0.025 |
| DBP | 0.987 | 0.981  0.992 | <0.001 | 0.995 | 0.996  1.000 | 0.038 |
| SBP | 0.995 | 0.982  0.992 | <0.001 | 0.994 | 0.988  1.000 | 0.071 |
| Rerular hct level | Reference <0.001 | | | | | |
| Low hct level | 1.369 | 1.116  1.745 | 0.003 | 1.589 | 1.099  2.297 | <0.001 |
| High hct level | 1.155 | 0.908  1.470 | 0.243 | 1.330 | 1.007  1.757 | 0.055 |
| RDW | 1.120 | 1.091  1.149 | <0.001 | 1.072 | 1.040  1.105 | <0.001 |
| Potassium | 1.078 | 0.998  1.165 | 0.057 | | | |
| Scr | 1.066 | 1.027  1.105 | 0.001 | 1.014 | 0.971  1.059 | 0.519 |
| Albumin | 0.915 | 0.841  0.994 | 0.037 | 0.941 | 0.859  1.031 | 0.191 |
| Hemoglobin | 0.986 | 0.929  1.009 | 0.121 | | | |
| Platelet | 1.000 | 0.998  1.000 | <0.001 | 0.999 | 0.999  1.000 | 0.034 |
| Bilirubin | 1.035 | 1.026  1.045 | <0.001 | 1.026 | 1.014  1.038 | <0.00 |
| Lactate | 1.123 | 1.099  1.148 | <0.001 | 1.087 | 1.061  1.114 | <0.001 |
| SOFA | 1.002 | 0.985  1.020 | 0.789 | | | |
| SAPSII | 1.020 | 1.016  1.024 | <0.001 | 1.011 | 1.006  1.016 | <0.001 |
| Diabetes | 1.014 | 0.754  1.365 | 0.952 | | | |
| Hypertension | 0.788 | 0.643  0.965 | 0.021 | 0.627 | 0.507  0.775 | <0.001 |
| COPD | 1.109 | 0.958  1.478 | 0.116 | | | |
| Respiratory failure | 0.950 | 0.831  1.086 | 0.455 | | | |
| CHF | 1.666 | 1.221  2.273 | 0.001 | 1.357 | 0.986  1.866 | 0.061 |
| Mechanical ventilation | 1.981 | 1.703  2.305 | <0.001 | 0.460 | 0.393  0.540 | <0.001 |

hct, hematocrit; COPD, Chronic obstructive pulmonary disease; CHF, Congestive heart failure; SBP, Systolic blood pressure; DBP, Diastolic blood pressure; MAP, mean arterial pressure; RDW, Red blood cell distribution width; Scr, Serum creatinine; SOFA, Sequential Organ Failure Assessment; SAPSII, Simplified Acute Physiology Scores II.

death in the high hct level also increased, the difference was not statistically significant (P = 0.055) (Table 3).

## Kaplan-Meier survival curve analysis

The Kaplan-Meier survival curve was drawn according to the category of hct to show the 30-day survival rate of patients with sepsis. The results showed that the difference between HCT level and the 30-day mortality rate of sepsis was statistically significant (P = 0.040), red blood cell ratio Content is related to the prognosis of patients with sepsis (Fig 1).

## Discussion

Recently, sepsis remains the main cause of death triggered by the infection in ICUs despite of considerable breakthroughs in the exploration of sepsis, including broad-spectrum antibiotics, supportive treatment, and even precision medicine and monitoring [10]. Infection and immune response disorders have been recognized as risk factors for organ dysfunctions. The poor functional status has been reported as a risk factor for sepsis and an expected consequence based on the long-term investigation of sepsis over the past decade [11, 12]. Therefore,

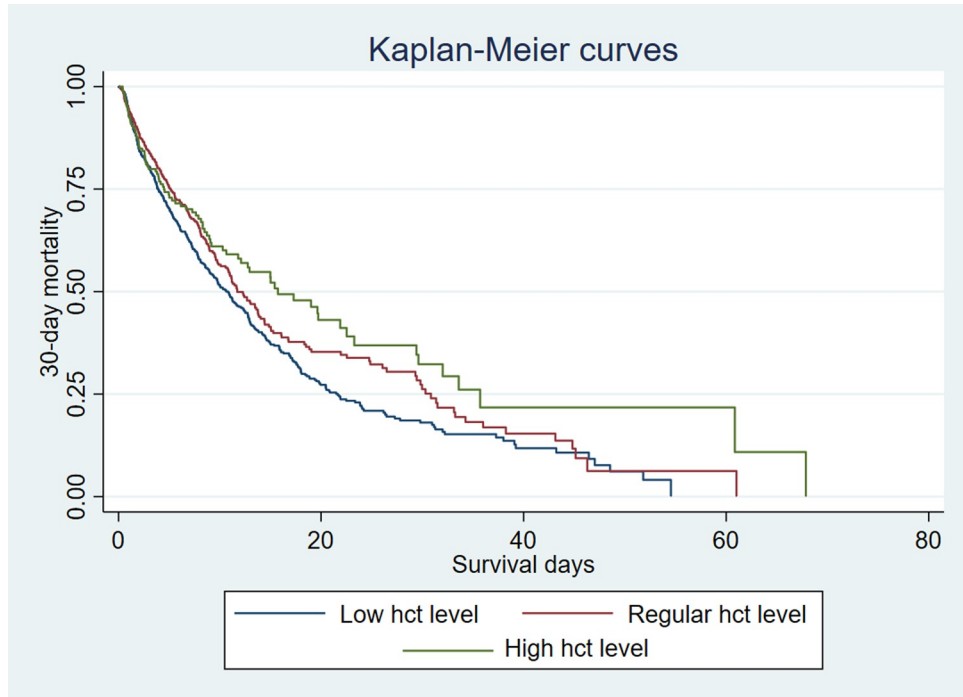

**Fig 1. Survival curves showing the association between the hct level and 30-day mortality.**

it is essential to be able to early predict and evaluate the prognostic factors for sepsis. Infection is one of the crucial reasons for the occurrence and development of sepsis. This is because it can touch the systemic inflammatory response and cause the formation of inflammasomes in the body, thereby inducing the release of numerous pro-inflammatory factors [13]. Therefore, it is imperative to use effective biomarkers to monitor the prognosis of sepsis. Whole blood parameters have been partially studied as biomarkers in the diagnosis, treatment, and prognosis of sepsis. Whole blood viscosity, red blood cell aggregation, and red blood cell deformability may be risk factors for sepsis and septic shock mortality [14]. In addition, the study conducted by Gong Yan et al. [15] has elucidated that elevated RDW can remarkably predict disease progression and poor clinical outcomes in patients with sepsis. Currently, several emerging studies have been conducted to ascertain the influence of HCT on the prognosis of sepsis. It has been elaborated that HCT is correlated with all-cause mortality in end-stage renal disease, heart failure, coronary heart disease, cancer, and inflammatory states [16]. However, HCT has not been determined as an independent influencing factor for clinical outcomes of diseases. In this study, 2057 patients with sepsis were evaluated to figure out the correlation between the HCT level and the prognosis of sepsis. The results illustrated that compared with septic patients with the regular HCT level, the 30-day mortality rate was enhanced in septic patients with a low HCT level measured within 24 hours after entering the ICU (male ≤ 42% and female ≤ 37%), accompanied by conspicuously augmented Simplified Acute Physiology Score II scores. Following gradual regression and correction of various potential confounding factors, we obtained reliable results from the study, suggesting that HCT might be an independent risk factor for the prognosis of patients with sepsis, which is the same as the previous group of sepsis anemia patients and Western countries. The results of a small-scale clinical trial are concordant [7, 17].

HCT is a whole blood parameter that reflects the ratio between red blood cells and plasmas. It is closely linked to the prognosis of critically ill patients and indicators for the fluid resuscitation treatment. However, the relationship between HCT and the prognosis of patients with sepsis remains poorly understood. Acute systemic infections lead to inflammatory responses that result in sepsis, which dramatically diminishes the number of red blood cells entering the blood circulation [18]. The production of reactive oxygen species may contribute to the repression of the ability of red blood cells to transport oxygen and the deformities of red blood cell membranes. During the entire process of inflammatory response and oxidative stress, the number of red blood cells is diminished and the blood dilution induced by liquid expansion leads to a reduction in the HCT level [19–21]. It is necessary to further prove these research hypotheses. It has been documented that inflammatory factors like tumor necrosis factor α, interleukin (IL)-1, IL-2, IL-6, and IL-8 trigger the adhesion of neutrophils and endothelial cells, leading to the formation of microthrombi [22]. This mechanism may be related to the rapid removal of red blood cells from the circulating blood caused by inflammation and oxidative stress in sepsis, which is manifested by anemia and accelerated red blood cell apoptosis [23]. In addition, red blood cell infusion has been highlighted to improve oxygen transport and metabolism, thereby alleviating microcirculation disorders in patients with sepsis and septic shock [24]. Therefore, the hypotheses about the HCT were proposed on the basis of the aforementioned research. As reflected by the results of this study, patients with the low HCT level exhibit low platelets, hemoglobin, and the high 30-day mortality rate. The mechanism mentioned above can provide a basis for this theory. Paolo Boffetta et al. [16] unraveled that HCT is also associated with the mortality of ordinary people. A retrospective study by Zhang Xin et al. [25] unveiled that the low HCT also correlated to the poor prognosis of patients with lung cancer and ovarian cancer. These findings suggest that HCT may reflect the severity of a wide range of diseases.

This study was a large-scale retrospective study, in which plenty of factors related to the death of sepsis were harvested and adjusted to evaluate their impacts on the prognosis. The results manifested that HCT could be an independent risk factor for the prognosis of patients with sepsis. Although HCT is not highly specific according to our results, it is a simple laboratory parameter that is easier to obtain. The HCT levels obtained in our research were all information of patients with sepsis on the first day after entering the ICU, minimizing the changes in the HCT levels caused by the progression of the disease and the treatment process. However, there also exist several limitations in this study: (1) The Medical Information Market Intensive Care Database is a sizeable single-center database that lacks diversification. Therefore, our results are influenced by unity and may have inevitable bias. (2) Due to the singularity of the database, we can only conduct observational researches on HCT and mortality, so further studies are needed to understand the underlying pathophysiological mechanism in the future. (3) Taking the problem of the content of the data sample into account, the sample is not classified by gender, but a general study is carried out.

In summary, HCT is associated with the prognosis of septic patients during admission to the ICU. Septic patients with the low HCT level presented with enhanced disease severity and high mortality rate. HCT can be applied as an independent risk factor for the prognosis of patients with sepsis.

Further study is required to investigate the pathophysiological and immunological mechanism of the relationship between HCT and clinical outcomes in septic patients. A perfect biomarker for sepsis has not yet been identified. More immunological experiments and multicenter studies on this easily available parameter will be of great significance for the early prediction of the outcomes of sepsis, which must be beneficial for the treatment of sepsis.

## Conclusion

In summary, the hematocrit in the first 24 hours after ICU admission was independently associated with increased 30-day all-cause mortality in adult septic patients but of limited sensibility and specificity. Further extensive multicenter prospective studies are needed to confirm the relationship and validate whose clinical significance.

## Author Contributions

**Conceptualization:** Mengdi Luo.

**Data curation:** Mengdi Luo, Na Li, He Qing.

**Formal analysis:** Mengdi Luo, He Qing.

**Methodology:** Yang Chen, Yuan Cheng.

**Writing – original draft:** Yang Chen, Yuan Cheng, Na Li.

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
