## [Decision Letter · Decision Letter 0]

14 Jan 2022

PONE-D-21-35992Association between hematocrit and the 30-day mortality of patients with sepsis: a retrospective analysis based on the large-scale clinical database MIMIC-IVPLOS ONE

Dear Dr. luo,

Thank you for submitting your manuscript to PLOS ONE. After careful consideration, we feel that it has merit but does not fully meet PLOS ONE’s publication criteria as it currently stands. Therefore, we invite you to submit a revised version of the manuscript that addresses the points raised during the review process.

 Two Reviewers experts in the field and on big data management reviewed the paper. I agree on their comments, especially about the methods used, which I ask you to fully solve

We look forward to receiving your revised manuscript.

Kind regards,

Andrea Cortegiani, M.D.

Academic Editor

PLOS ONE

Journal Requirements:

 (No)

 (No)

7. Please include your tables as part of your main manuscript and remove the individual files. Please note that supplementary tables (should remain/ be uploaded) as separate "supporting information" files.

8. Thank you for stating the following in the Funding Section of your manuscript: 

(This study was supported by (the Chengdu Science and Technology 

Bureau (No.2017FZ0058)) and (the Health Department of Sichuan Province 

(No.17PJ474)))

(No)

Reviewers' comments:

Reviewer's Responses to Questions

**Comments to the Author**

1. Is the manuscript technically sound, and do the data support the conclusions?

Reviewer #1: Yes

Reviewer #2: Yes

2. Has the statistical analysis been performed appropriately and rigorously? 

Reviewer #1: Yes

Reviewer #2: Yes

3. Have the authors made all data underlying the findings in their manuscript fully available?

Reviewer #1: Yes

Reviewer #2: Yes

4. Is the manuscript presented in an intelligible fashion and written in standard English?

Reviewer #1: Yes

Reviewer #2: Yes

5. Review Comments to the Author

Reviewer #1: Thank you for the opportunity to review this study.

The authors have conducted a retrospective observational study looking for an association between hematocrit and 30 day mortality in a septic population. The method proposed is a multivariate model of a single center retrospective dataset (MIMIC IV). They found a correlation between low HCT value at admission and increased 30d mortality.

The study finding are not very original. I have the following concerns and comments:

1) Introduction paragraph (line 81-93) seems repeated

2) reporting checklist is missing, I suggest to use the RECORD guidelines (The REporting of studies Conducted using Observational Routinely-collected health Data ) Please refer to http://journals.plos.org/plosmedicine/article?id=10.1371/journal.pmed.1001885

3) Please indicate in detail how do you selected the sepsis population in the MIMIC database. Did you used the "concepts" script for mimic-iv in order to improve the reproducibility of the analysis? (Johnson, A. E., Stone, D. J., Celi, L. A., & Pollard, T. J. (2018). The MIMIC Code Repository: enabling reproducibility in critical care research. Journal of the American Medical Informatics Association : JAMIA, 25(1), 32–39. or an extraction based on ICD-9 codes (Epidemiology of severe sepsis in the United States: analysis of incidence, outcome, and associated costs of care. Angus DC, Linde-Zwirble WT, Lidicker J, Clermont G, Carcillo J, Pinsky MRCrit Care Med. 2001 Jul; 29(7):1303-10.) and Martin et al. (The epidemiology of sepsis in the United States from 1979 through 2000. Martin GS, Mannino DM, Eaton S, Moss M

N Engl J Med. 2003 Apr 17; 348(16):1546-54)

4) Line 119-120 report the use of non parametric test, please specify the distribution of data

Reviewer #2: Comments for authors:

Dear authors,

You have presented a large-scale retrospective analysis on the association between hematocrit and mortality in patients with sepsis.

The topic is interesting and it would be important to find a real contribution in this setting.

Please give your statement to the following points:

1. Abstract

The abstract is clear enough.

2. Introduction

- The introduction was repeated twice; delete the second repetition.

3. Materials and Methods

- It has not been included if a sample size estimation has been carried out; has it been calculated?

- It is not clear enough to me whether the selected patients were already diagnosed with sepsis upon entering the ICU.

4.Results

- In the figure 1, please add legend.

5. Discussion

- It is written that "The HCT levels obtained in our research were all information of patients with sepsis on the first day after entering the ICU, which suppressed the progression of the disease and changes in laboratory parameter values induced by fluid resuscitation therapy"; but the patients who had already been diagnosed with sepsis upon admission to the ICU could not have already started fluid resuscitation therapy? Or could not they have been more fragile patients already in fluid overload?

- Please specify better the clinical message that the authors want to send.

6. Conclusion

- In my opinion the words "hematocrit during ICU" is misleading; better to specify “hematocrit in the first 24 hours after ICU admission”.

7. Tables

- Table 2 appears to be repeated; please check the data.

- Tables 1 and 2 lack legends; please enter.

8. References

Please check the journal’s guidelines

It would be necessary to revisit the English language.

Best regards

6. PLOS authors have the option to publish the peer review history of their article (what does this mean?). If published, this will include your full peer review and any attached files.

Reviewer #1: **Yes: **Alberto Noto

Reviewer #2: No

---

## [Author Response · Author response to Decision Letter 0]

10 Feb 2022

Dear editors and reviewers:

Thanks for your letter and the reviewers' comments about our manuscript entitled “Association between hematocrit and the 30-day mortality of patients with sepsis: a retrospective analysis based on the large-scale clinical database MIMIC-IV”.All these suggestions are valuable for me to revise my manuscript. We have studied comments carefully and have made corrections which we hope meet with approval.

Suggestions from editors:

The topic is interesting and it would be important to find a real contribution in this setting.

Please give your statement to the following points: 

1. Abstract

The abstract is clear enough.

2. Introduction

- The introduction was repeated twice; delete the second repetition.

3. Materials and Methods

- It has not been included if a sample size estimation has been carried out; has it been calculated?

- It is not clear enough to me whether the selected patients were already diagnosed with sepsis upon entering the ICU.

4.Results

- In the figure 1, please add legend. 

5. Discussion

- It is written that "The HCT levels obtained in our research were all information of patients with sepsis on the first day after entering the ICU, which suppressed the progression of the disease and changes in laboratory parameter values induced by fluid resuscitation therapy"; but the patients who had already been diagnosed with sepsis upon admission to the ICU could not have already started fluid resuscitation therapy? Or could not they have been more fragile patients already in fluid overload?

- Please specify better the clinical message that the authors want to send.

6. Conclusion

- In my opinion the words "hematocrit during ICU" is misleading; better to specify “hematocrit in the first 24 hours after ICU admission”.

7. Tables 

- Table 2 appears to be repeated; please check the data. 

- Tables 1 and 2 lack legends; please enter.

8. References 

Please check the journal’s guidelines

It would be necessary to revisit the English language. 

Dear editors,the revised portion are marked in red in our paper.

Responses to Reviwers:

Reviewer #1

1. -Introduction paragraph (line 81-93) seems repeated

We are very sorry for our neligence of this paragraph.And then the repetitive sentence in the introductory paragraph has been deleted.

2. -reporting checklist is missing, I suggest to use the RECORD guidelines (The REporting of studies Conducted using Observational Routinely-collected health Data ) Please refer to http://journals.plos.org/plosmedicine/article?id=10.1371/journal.pmed.100188-

Considering the reviwer’s suggestion,we have made some modifications based on the guidelines.

3.-Please indicate in detail how do you selected the sepsis population in the MIMIC database. Did you used the "concepts" script for mimic-iv in order to improve the reproducibility of the analysis? (Johnson, A. E., Stone, D. J., Celi, L. A., & Pollard, T. J. (2018). The MIMIC Code Repository: enabling reproducibility in critical care research. Journal of the American Medical Informatics Association : JAMIA, 25(1), 32–39. or an extraction based on ICD-9 codes (Epidemiology of severe sepsis in the United States: analysis of incidence, outcome, and associated costs of care. Angus DC, Linde-Zwirble WT, Lidicker J, Clermont G, Carcillo J, Pinsky MRCrit Care Med. 2001 Jul; 29(7):1303-10.) and Martin et al. (The epidemiology of sepsis in the United States from 1979 through 2000. Martin GS, Mannino DM, Eaton S, Moss M

N Engl J Med. 2003 Apr 17; 348(16):1546-54)

Firstly,we download the zip package of the database from the official website for a series of installation, and then enter the interface of PostgreSQL 6.0. We use SELECT*FROM statement to find the icd-9 code of sepsis in d_icd_diagnoses, and then use SELECT WHERE statement to find the related icd-9 code of sepsis in diagnoses_icd (99591/99592)We used the "concepts" script for mimic-iv in order to improve the reproducibility of the analysis.

4.-Line 119-120 report the use of non parametric test, please specify the distribution of data

Considering the reviewer’s suggestion the data distribution has been shown in Table 1.

Reviewer #2

1.Introduction

- The introduction was repeated twice; delete the second repetition.

Thank you for your careful review.The repetitive sentence in the introductory paragraph has been deleted.

2.Materials and Methods

- It has not been included if a sample size estimation has been carried out; has it been calculated?

- It is not clear enough to me whether the selected patients were already diagnosed with sepsis upon entering the ICU.

This sample size has been calculated.The data of sepsis patients we present in the MIMIC IV database by SQL statements are diagnosed on admission to the ICU.This can be determined by referring to the r

elevant details of this database on the official website.(https://physionet.org/content/mimiciv/0.4/).

3.Results

- In the figure 1, please add legend.

Thank you for your careful review.We have added the legend.(Details can be found in the revised version)

4.Discussion

- It is written that "The HCT levels obtained in our research were all information of patients with sepsis on the first day after entering the ICU, which suppressed the progression of the disease and changes in laboratory parameter values induced by fluid resuscitation therapy"; but the patients who had already been diagnosed with sepsis upon admission to the ICU could not have already started fluid resuscitation therapy? Or could not they have been more fragile patients already in fluid overload?

Thank you for the reviewer's suggestion, We have reworked it and marked in red.

5.Conclusion

- In my opinion the words "hematocrit during ICU" is misleading; better to specify “hematocrit in the first 24 hours after ICU admission”.

Thank you for the reviewer's suggestion, I have adopted your revision and made the statement changes.

6.Tables

- Table 2 appears to be repeated; please check the data.

- Tables 1 and 2 lack legends; please enter.

The table has been made in detail。（Details can be found in the revised version）

7.References

Please check the journal’s guidelines

It would be necessary to revisit the English language.

Changes have been made to the reference specification.

Thank you for your careful review. We really appreciate your efforts in reviewing our manuscript during this unprecedented and challenging time. We wish good health to you, your family, and community. Your careful review has helped to make our study clearer and more comprehensive.

---

## [Decision Letter · Decision Letter 1]

8 Mar 2022

Association between hematocrit and the 30-day mortality of patients with sepsis: a retrospective analysis based on the large-scale clinical database MIMIC-IV

PONE-D-21-35992R1

Dear Dr. luo,

We’re pleased to inform you that your manuscript has been judged scientifically suitable for publication and will be formally accepted for publication once it meets all outstanding technical requirements.

Kind regards,

Andrea Cortegiani, M.D.

Academic Editor

PLOS ONE

Additional Editor Comments (optional):

Reviewers' comments:

Reviewer's Responses to Questions

**Comments to the Author**

1. If the authors have adequately addressed your comments raised in a previous round of review and you feel that this manuscript is now acceptable for publication, you may indicate that here to bypass the “Comments to the Author” section, enter your conflict of interest statement in the “Confidential to Editor” section, and submit your "Accept" recommendation.

Reviewer #1: All comments have been addressed

2. Is the manuscript technically sound, and do the data support the conclusions?

Reviewer #1: Yes

3. Has the statistical analysis been performed appropriately and rigorously? 

Reviewer #1: Yes

4. Have the authors made all data underlying the findings in their manuscript fully available?

Reviewer #1: Yes

5. Is the manuscript presented in an intelligible fashion and written in standard English?

Reviewer #1: Yes

6. Review Comments to the Author

Reviewer #1: (No Response)

7. PLOS authors have the option to publish the peer review history of their article (what does this mean?). If published, this will include your full peer review and any attached files.

Reviewer #1: **Yes: **Alberto Noto

---

## [Editor Report · Acceptance letter]

15 Mar 2022

PONE-D-21-35992R1 

Association between hematocrit and the 30-day mortality of patients with sepsis: a retrospective analysis based on the large-scale clinical database MIMIC-IV 

Dear Dr. luo:

I'm pleased to inform you that your manuscript has been deemed suitable for publication in PLOS ONE. Congratulations! Your manuscript is now with our production department. 

Kind regards, 

on behalf of

Dr. Andrea Cortegiani 

Academic Editor

PLOS ONE